# Rosmarinic Acid, as an NHE1 Activator, Decreases Skin Surface pH and Improves the Skin Barrier Function

**DOI:** 10.3390/ijms23073910

**Published:** 2022-03-31

**Authors:** Seung-Won Jung, Gi Hyun Park, Eunjung Kim, Kang Min Yoo, Hea Won Kim, Jin Soo Lee, Min Youl Chang, Kyong-Oh Shin, Kyungho Park, Eung Ho Choi

**Affiliations:** 1Department of Dermatology, Yonsei University Wonju College of Medicine, Wonju 26426, Korea; seungwon0826@naver.com (S.-W.J.); hyksuj0326@naver.com (E.K.); 2CMS LAB, Seoul 35324, Korea; park0503@wonik.com (G.H.P.); yookm@wonik.com (K.M.Y.); khw1225@wonik.com (H.W.K.); sooleevn@wonik.com (J.S.L.); 3SKINMED Clinical Trials Center, Daejeon 25437, Korea; mychang195@gmail.com; 4Department of Food Science and Nutrition, Convergence Program of Material Science for Medicine and Pharmaceutics, Hallym University, Chuncheon 24252, Korea; 0194768809@hanmail.net (K.-O.S.); hopark78@gmail.com (K.P.)

**Keywords:** skin barrier, skin pH, sodium proton exchanger 1, *Melissa officinalis* leaf extract, rosmarinic acid

## Abstract

Stratum corneum (SC) pH regulates skin barrier functions and elevated SC pH is an important factor in various inflammatory skin diseases. Acidic topical formulas have emerged as treatments for impaired skin barriers. Sodium proton exchanger 1 (NHE1) is an important factor in SC acidification. We investigated whether topical applications containing an NHE1 activator could improve skin barrier functions. We screened plant extracts to identify NHE1 activators in vitro and found *Melissa officinalis* leaf extract. Rosmarinic acid, a component of *Melissa officinalis* leaf extract, significantly increased NHE1 mRNA expression levels and NHE1 production. Immunofluorescence staining of NHE1 in 3D-cultured skin revealed greater upregulation of NHE1 expression by NHE1 activator cream, compared to vehicle cream. Epidermal lipid analysis revealed that the ceramide level was significantly higher upon application of the NHE1 activator cream on 3D-cultured skin, compared to application of a vehicle cream. In a clinical study of 50–60-year-old adult females (*n* = 21), application of the NHE1 activator-containing cream significantly improved skin barrier functions by reducing skin surface pH and transepidermal water loss and increasing skin hydration, compared to patients who applied vehicle cream and those receiving no treatment. Thus, creams containing NHE1 activators, such as rosmarinic acid, could help maintain or recover skin barrier functions.

## 1. Introduction

The human skin surface has a physiologically acidic pH, and many studies have postulated that the natural skin surface pH is below 5 [1,2]. In the stratum corneum (SC), the pH ranges from approximately 5 in the outermost layer to approximately neutral pH values in the deeper layer [3]. This acidic surface pH and pH gradient are important for the maintenance of skin health. Alterations in SC pH result in functional abnormalities of the skin barrier. Skin pH increases in inflammatory skin diseases, including atopic eczema, psoriasis, acne, and dry skin, and in both neonatal and in aged skin [4,5,6,7,8]. Therefore, weakly acidic topical formulas have recently emerged as treatment strategies to decrease skin pH in such cases.

The SC pH regulates barrier functions, such as SC permeability homeostasis and SC integrity/cohesion [9,10]. SC neutralization caused by exposure to neutral pH buffer or super bases induces functional abnormalities, leading to delayed permeability barrier recovery and decreased SC integrity/cohesion [11,12]. These pH-induced barrier abnormalities are associated with pH-dependent enzymes in SCs. Lipid-processing enzymes, such as β-glucocerebrosidase and acidic sphingomyelinase, exhibit acidic pH optima (pH 4.5–5) [13,14,15] and form lamellar bilayers by generating ceramides from glucosylceramide and sphingomyelin precursors, respectively [14,16]. Serine proteases, especially stratum corneum tryptic enzyme (SCTE) and stratum corneum chymotryptic enzyme (SCCE), which exhibit neutral pH optima [17,18], degrade the extracellular protein structure of corneodesmosomes, such as desmoglein-1 (DSG1), desmocollin-1 (DSC1), and corneodesmosin (CDSN) [19]. Therefore, elevated pH inhibits lipid-processing enzymes and activates serine proteases, thereby damaging the SC barrier function and SC integrity/cohesion.

Acidification of SC is not dependent on one factor but is regulated by both endogenous and exogenous mechanisms [10,20]. Endogenous mechanisms, such as the histidine-to-urocanic acid pathway, the phospholipid-to-FFA pathway, and sodium proton exchanger 1 (NHE1) have been shown to both affect SC pH and regulate one or more SC functions. They interact to maintain an acidic pH because a decrease in the level of urocanic acid (UCA) or pyrrolidone carboxylic acid (PCA) in filaggrin deficiency is accompanied by an increase in NHE1 and secretory phospholipase A2 (sPLA2) levels [21]. 

NHE1 controls acidification of the extracellular domain in the stratum granulosum (SG)/SC interface, which is essential for the activation of two lipid-processing enzymes and the formation of a permeability barrier [21]. NHE1 inactivation is sufficient to impede SC acidification, and specifically, alter the pH at the SG/SC interface and skin surface [22]. The barrier function of skin decreases with age, and the skin exhibits decreased SC hydration and elevated surface pH [23,24]. NHE1 expression levels decrease in moderately aged skin, after which SC pH decreases [4], which results in delayed barrier recovery due to defective extracellular lipid processing [4,22]. Furthermore, neutralizing SC pH by acute barrier disruption upregulates NHE1 expression within hours [25]. In this context, topical applications containing an NHE1 activator may be more effective than simple acidic topical applications. We screened plant extracts to identify activators of NHE1 in vitro. Next, we investigated whether rosmarinic acid activates NHE1; decreases skin surface pH; and increases levels of SC intercellular lipids, such as ceramide, cholesterol, and fatty acids, in 3D-cultured skin. Finally, we evaluated whether plant extracts with rosmarinic acid are clinically effective.

## 2. Results

### 2.1. Melissa Officinalis Leaf Extract and Rosmarinic Acid Promote NHE1 Expression In Vitro

To identify the activator of NHE1 expression in vitro, we screened various plant extracts and then found *M. officinalis* leaf extract. The test solution containing *M. officinalis* leaf extract significantly increased the NHE1/GAPDH ratio (i.e., the ratio between the *NHE1* and *GAPDH* expression levels) compared to the control solution (0.05% solution, 124.0 ± 1.3%, *p* < 0.001; 0.1% solution, 127.6 ± 4.1%, *p* < 0.01) (Figure 1a). In particular, the increase in NHE1 mRNA expression level was proportional to the concentration of *M. officinalis* leaf extract. *M. officinalis* leaf extract solution significantly increased the amount of NHE1 per total protein, compared to the control solution (0.05% solution, 112.7 ± 3.0%, *p* < 0.05; 0.1% solution, 116.4 ± 1.1%, *p* < 0.01) (Figure 1b), and this effect was proportional to the *M. officinalis* leaf extract concentration.

We confirmed that *M. officinalis* leaf extract acts as an NHE1 activator, even though it contains various other components. As rosmarinic acid is known to be the most predominant compound among them [26,27], experiments on the promoting effect of rosmarinic acid on NHE1 mRNA and NHE1 expression were conducted in vitro.

Rosmarinic acid (4 µg/mL) solution significantly increased the NHE1/GAPDH ratio (134.6 ± 5.6%) compared to the control solution (*p* < 0.01), and although the result was not statistically significant, rosmarinic acid (2 µg/mL solution) increased NHE1 mRNA expression level (118.3 ± 8.2%), compared to the control solution (*p* > 0.05) (Figure 2a).

Similar to *M. officinalis* leaf extract, rosmarinic acid increased NHE1 mRNA expression in a concentration-dependent manner. Additionally, rosmarinic acid solution significantly increased the amount of NHE1 per total protein, in a concentration-dependent manner, compared to the control solution (2 µg/mL solution, 133.2 ± 2.3%, *p* < 0.001; 4 µg/mL solution, 136.9 ± 11.3%, *p* < 0.01) (Figure 2b).

### 2.2. Rosmarinic Acid Increases Immunofluorescence of NHE1 and the Ceramide Level 

#### 2.2.1. Immunofluorescence Staining (IF) for Analyzing NHE1 Expression in 3D-Cultured Skin

IF of 3D-cultured skin was performed to confirm that rosmarinic acid activates epidermal NHE1. After the application of the NHE1 activator creams containing 0.05% or 0.1% rosmarinic acid or vehicle cream, NHE1 expression in the SC and SG was compared (Figure 3). The NHE1 expression level was increased by NHE1 activator creams compared to the vehicle cream, and the increase in the NHE1 level was proportional to the rosmarinic acid concentration (Figure 3b–d). Furthermore, at the same rosmarinic acid concentration, the NHE1 expression level was proportional to the duration of the NHE1 activator cream application (Figure 3c–h).

#### 2.2.2. Epidermal Lipid Analysis in 3D-Cultured Skin

After the application of the topical NHE1 activator cream (containing 0.05% or 0.1% rosmarinic acid) or the vehicle cream, the levels of ceramide NS, cholesterol, and fatty acids were analyzed after 48 h. The total ceramide content was significantly higher after application of the NHE1 activator cream compared to the control (0.05% rosmarinic acid cream vs. vehicle cream, *p* < 0.05; and 0.1% rosmarinic acid cream vs. vehicle cream; *p* < 0.01) (Figure 4a). In particular, there was no significant difference based on ceramide chain length, and although the difference was insignificant, 0.1% rosmarinic acid cream increased total ceramide content compared to 0.05% cream (*p* = 0.13). However, there were no significant differences in the amounts of total fatty acid and cholesterol according to chain length (Figure 4b) and cholesterol (Figure 4c).

### 2.3. Application of the NHE1 Activator-Containing Skin Care Cream for 4 Weeks Lowers Skin Surface pH

Skin pH changes and skin pH recovery after sodium lauryl sulphate (SLS) irritation were assessed in aged women after application of the NHE1 activator-containing skin care cream (*M. officinalis* leaf extract 0.05% and rosmarinic acid 0.1%) on the forearm, and application of the vehicle cream or no application on the other forearm.

After topical application of the NHE1 activator-containing cream, skin pH decreased significantly after 4 weeks (pH 5.15 at week 0 vs. 4.83 at week 4; *p* < 0.05). Application of the vehicle cream resulted in an increase in skin pH after 4 weeks, but the difference was not statistically significant (pH 5.18 at week 0 vs. 5.29 at week 4, *p* = 0.701). There was a significant difference in pH between the NHE1 activator cream sites and vehicle cream sites (*p* < 0.01), and between the NHE1 activator cream sites and untreated sites (*p* < 0.001) at week 4 (Figure 5a).

After topical application (NHE1 activator cream or vehicle cream) or no treatment for 4 weeks, damage was induced at the same sites on each forearm using a patch containing 1% SLS for 24 h. Skin pH recovery was measured over time, on days 0, 1, 3, and 7, and cream was not applied after SLS irritation. Seven days after SLS irritation, the skin pH decreased on day 7 at the sites where the NHE1 activator cream was applied (day 0, 5.19 vs. day 3, 5.24, *p* = 0.450 and day 7, 5.02; *p* < 0.01) (Figure 5b). However, the skin pH increased until day 7 after SLS irritation at the vehicle cream application site (day 0, 5.51 vs. day 3, 5.66, *p* < 0.001 and day 7, 5.55, *p* < 0.001) and the untreated skin (day 0, 5.34 vs. day 3, 5.50, *p* = 0.172 and day 7, 5.40; *p* < 0.05).

### 2.4. Application of the NHE1 Activator-Containing Skin Care Cream Reduces Transepidermal Water Loss (TEWL) and Improves Skin Hydration

The functional parameters of the skin barrier, such as TEWL and skin hydration, were evaluated after application of cream containing the NHE1 activator (*M. officinalis* leaf extract 0.05% and rosmarinic acid 0.1%) on one forearm, and the vehicle cream on the other forearm.

After topical application of the NHE1 activator-containing skin care cream, TEWL decreased significantly after weeks 2 and 4 (week 0, 12.1 vs. week 2, 11.7, *p* = 0.003 and week 4, 10.7, *p* < 0.001), while TEWL in the vehicle cream remained at the baseline level (week 0, 13.0 vs. week 2, 13.2, *p* = 0.527 and week 4, 12.8, *p* = 0.526). At week 4, there was a significant difference in TEWL in skin treated with the NHE1 activator-containing cream compared to that treated with the vehicle cream (week 4 after application of the NHE1 activator cream, 10.7 vs. vehicle cream, 12.8; *p* = 0.028) and untreated skin (week 4 after application of the NHE1 activator cream, 10.7 vs. no treatment sites, 12.4; *p* < 0.001). (Figure 6a).

Skin hydration increased significantly after application of the NHE1 activator cream (week 0 after application, 38.9; week 2 after application, 49.8, *p* < 0.001; week 4 after application, 58.1, *p* < 0.001) and vehicle cream (week 0, 36.9; week 2, 41.2, *p* < 0.001; week 4, 47.2, *p* < 0.001) (Figure 6b). The NHE1 activator cream also showed significantly higher skin hydration than the vehicle cream at weeks 2 and 4 (week 2; *p* < 0.001, and week 4; *p* < 0.001).

## 3. Discussion

Aged skin is known to exhibit frequent irritant contact dermatitis, xerosis, and poor wound healing. This could be due to abnormal skin barrier function [28]. It has been reported that aged skin can be divided into fully aged (> 75 years) and moderately aged (55–74 years) skin and decreased epidermal lipid synthesis and secretion are the primary defects of the abnormal barrier in fully aged skin [29,30]. However, delayed lipid processing due to defective acidification is considered a primary defect rather than decreased lipid synthesis and secretion in moderately aged skin [4]. The elevation of SC pH increases the activity of serine proteases (SPs), which results in the degradation of corneodesmosomes and abnormal SC integrity, and defective SC acidification weakens skin barrier function [31].

In this study, we found that creams containing rosmarinic acid increased NHE1 expression levels compared to a vehicle cream in 3D-cultured skin, and elevated NHE1 expression positively correlated with rosmarinic acid concentration and duration of cream application (Figure 3). Results from our 3D skin culture experiments validate our hypothesis that rosmarinic acid can activate NHE1. In particular, the application of cream containing rosmarinic acid reportedly increased ceramide levels compared to the vehicle cream, as per the epidermal lipid analysis of 3D-cultured skin; however, there was no significant difference in fatty acid and cholesterol levels (Figure 4). These results are concordant with those of previous studies. Ceramide is the most important lipid for the skin barrier function, accounting for more than 50% of intercellular lipids in the SC. To generate functional ceramides from precursors, such as glucosylceramide and sphingomyelin, β-glucocerebrosidase and acidic sphingomyelinase are important respectively, and these ceramide-generating enzymes are activated when the acidity of the SC is maintained [32]. SC acidification by creams containing rosmarinic acid contributes to activation of ceramide-generating enzymes and ceramide formation and improves the skin barrier function in combination with various mechanisms, such as preventing CD degradation due to SP activation.

In our clinical study, application of the NHE1 activator-containing skin care cream significantly improved the skin barrier function, as indicated by decreased skin surface pH, lower TEWL, and elevated skin hydration compared to vehicle-treated and untreated controls (Figure 5 and Figure 6). In a previous study, abnormal SC acidification caused multiple defects in skin barrier functions, such as delayed permeability, barrier recovery, and dysregulated lipid processing, but barrier recovery was effectively normalized by reacidification [4]. Other previous studies have identified the critical role of NHE1 in the skin barrier function as NHE1 inhibition impaired skin barrier development or recovery and delayed lipid processing [12,22,25]. Therefore, we believe that application of the NHE1 activator-containing skin care cream can effectively enhance epidermal function in various ways.

As abnormal SC acidification in moderately aged skin does not improve with progression of aging, multiple defects related to abnormal SC acidification, in addition to defective lipid synthesis and secretion, deteriorate skin barrier functions in fully aged skin [4]. Therefore, rosmarinic acid-containing skin care creams are expected to be effective in all age groups, including fully-aged patients.

## 4. Materials and Methods

### 4.1. In Vitro Study

#### 4.1.1. Formulation of the Test Solutions

*M. officinalis* leaf extract (Maruzen Pharmaceuticals Co., Ltd., Hiroshima, Japan) containing RA at more than 0.03% was mixed in a keratinocyte basal medium (KBM) at concentrations of 0.05% and 0.1%. Rosmarinic acid (Fujifilm Wako Pure Chemical Corporation, Tokyo, Japan) was dissolved in dimethyl sulfoxide (DMSO) and mixed in KBM medium at concentrations of 2 µg/mL and 4 µg/mL.

#### 4.1.2. Evaluation of Effects of the Test Solutions on NHE1 mRNA Expression

Normal human epidermal keratinocytes (NHEKs) were cultured in keratinocyte growth medium (KGM) in a 75 cm^2^ flask. Cells were collected by trypsin treatment and diluted with KGM to obtain a final density of 1.5 × 10^5^ cells/mL; then, 2 mL of the cell mixture was inoculated onto a 35-mm diameter dish and cultured overnight. KGM was then replaced with KBM, and the cells were cultured for 24 h. A test sample was dissolved in KBM and added to the dish. After 24 h of incubation, total RNA was isolated according to the standard method. Total RNA obtained from cells incubated without the test sample was isolated in the same manner. cDNA was synthesized by reverse transcription of 200 ng total RNA, and then, real-time PCR analyses were performed using a Thermal Cycler Dice^®^ Real Time System III (Takara Bio Inc., Otsu, Japan), TaKaRa SYBR^®^ PrimeScript^TM^ RT-PCR Kit (Perfect Real Time) (Takara Bio Inc., Otsu, Japan), and *NHE1* or glyceraldehyde-3-phosphate (*GAPDH*) primer set. Relative mRNA levels were expressed as the NHE1/GAPDH ratio.

The following formula was used to assess the effect of the sample on NHE1 mRNA expression:Promoting rate on NHE1 mRNA expression (%) = (A/B) × 100

A: NHE1/GAPDH ratio from a test sample.

B: NHE1/GAPDH ratio from control.

#### 4.1.3. Evaluation of the Effect of the Test Sample on NHE1 Production

After 48 h of incubation, the test sample dissolved in KBM was added to the well, and the cells were lysed with 400 µL of 50 mmol/L Tris-HCl buffer (pH 7.5), containing 150 mmol/L NaCl, 1% Nonidet P40, 0.5% sodium deoxycholate, and protease inhibitors, and then centrifuged to remove cell debris. The amount of NHE1 protein in 100 µL of the protein solution was measured using the NHE1 ELISA Kit (Cloud-Clone Corp., Houston, TX, USA). Simultaneously, total protein in 20 µL of the protein solution was quantified, and the amount of NHE1 per total protein was calculated. The amount of NHE1 per total protein extracted from untreated cells was regarded as 100, and the promoting effect on NHE1 production per total protein extracted from cells treated with the test sample was evaluated using the following formula:Promoting rate on NHE1 production (%) = (A/B) × 100

A: amount of NHE1 per total protein extracted from cells using a test sample.

B: amount of NHE1 per total protein extracted from cells without the test sample.

### 4.2. 3D-Cultured Skin

#### 4.2.1. Formulation of the NHE1 Activator Cream

The vehicle cream was formulated as an oil-in-water (O/W) cream consisting of water, glycerin, caprylic/capric triglyceride, cetearyl alcohol, dimethicone, 1,2-hexanediol, cetearyl olivate, sorbitan olivate, carbomer, tromethamine, ethylhexylglycerin, and citric acid to adjust the pH to 7.0. NHE1 activator cream for 3D-cultured skin was formulated by mixing 0.05% or 0.1% rosmarinic acid (Avention, Incheon, Korea) with vehicle cream.

#### 4.2.2. Immunofluorescent Analysis of NHE1 in 3D-Cultured Skin

The 3D-cultured skin (EpiDerm^TM^) was purchased from MatTek Life Science (Ashland, MA, USA), and rosmarinic acid (0.1% and 0.05%) cream and vehicle cream were applied to the 3D-cultured skin twice at an interval of 4 h. After 0, 12, 24, and 48 h of cream application, the 3D skin samples were washed with phosphate buffered saline (PBS) and fixed in 4% paraformaldehyde fixation, after which the skin was embedded in paraffin blocks for immunofluorescence analysis. After paraffin blocks of samples were prepared and sectioned, the samples were washed sequentially in xylene for 15 min, 100% ethanol for 10 min, 95% ethanol for 3 min, 70% ethanol for 1 min, and then boiled in citrate buffer. After washing with PBS for 15 min and blocking with Ultra V block buffer (Thermo Fisher Scientific, Waltham, MA, USA) for 5 min, the slides were incubated in primary antibody against NHE1 (dilution, 1:50; Bioclone, San Diego, CA, USA) at 4 °C overnight, followed by incubation in secondary anti-rabbit IgG (dilution, 1:100; Alex Fluor 488, Cell Signaling, Danvers, MA, USA) for 1 h. These slides were visualized by staining with diluted DAPI (1 mg/0.5 µL stock; Abcam, Cambridge, UK) and then covered with Mounting (lmmu-Mount™, Fisher Scientific, Hampton, NH, USA). Immunofluorescence images were obtained by observing the slides at 200× and 400× magnifications using a Zeiss laser scanning confocal microscope (Thornwood, NY, USA).

#### 4.2.3. Epidermal Lipid Analysis in 3D-Cultured Skin

The 0.1% and 0.05% rosmarinic acid and vehicle cream were applied twice, at an interval of 4 h, to the 3D-cultured skin (MatTek Life Science, Ashland, MA, USA) samples. Forty-eight hours after application of the cream, the samples were washed with PBS, placed in a 1.5 mL tube, and stored in a deep freezer (−80 °C) prior to lipid extraction using RIPA buffer, as reported previously [33,34]. The extracted lipids were dried using a vacuum system (Vision, Seoul, South Korea), re-dissolved in methanol, and analyzed using liquid chromatography-electrospray ionization tandem mass spectrometry (LC-ESI-MS/MS; API 5500 QTRAP mass, AB/SCIEX, Framingham, MA, USA) in the selected ion monitoring mode. The ceramide NS MS/MS transitions (*m*/*z*) were 510→264 for C14-ceramide NS, 538→264 for C6-ceramide NS, 566→264 for C18-ceramide NS, 594→264 for C20-ceramide NS, 622→264 for C22-ceramide NS, 648→264 for C24:1-ceramide NS, 650→264 for C24-ceramide NS, 676→264 for C26:1-ceramide NS, and 678→264 for C26-ceramide NS; the MS/MS transition was 552→264 for the d17:1/18:0 internal standard.

To quantify the free cholesterol and free fatty acid (FA) content, lipid extraction was performed using the Folch method with minor modifications [35,36,37]. Briefly, 3D skin was lysed and sonicated in a methanol/chloroform solution (1:2, *v*/*v*) containing butylated hydroxytoluene (500 µg/mL), followed by the addition of 500 pmol of docosahexaenoic acid and d_6_-cholesterol as an internal standard. The extracted lipids were dried using a vacuum system, redissolved in methanol, and analyzed via LC-ESI-MS/MS using the selective ion monitoring mode. First, both free cholesterol and FAs were separated by reverse-phase HPLC (ExionLC™ Series UHPLC, AB/SCIEX, Framingham, MA, USA) using a KINETEX C18 column (2.1 mm × 50 mm, ID: 2.6 µm) (Phenomenex, St. Louis, MO, USA), as described previously [37,38]. The *m*/*z* of the FAs were 227→183 for C14:0 FA, 253→209 for C16:1 FA, 255→211 for C16:0 FA, 277→233 for C18:3 FA, 279→235 for C18:2 FA, 281→237 C18:1 FA, 283→239 C18:0 FA, 303→259 for C20:4 FA, 311→267 for C20:0 FA, 337→293 for C22:1 FA, 339→295 for C22:0 FA, 365→321 for C24:1 FA, and 367→323 for C24 FA. The *m*/*z* of cholesterol were 369.3→161.5 or 369.3→147.1 for free cholesterol and 374.4→152.7 for d6-cholesterol. All data were acquired using Analyst 1.7.1 software (Applied Biosystems, Foster City, CA, USA). The details regarding our validation analysis are included in the online Appendix A.

### 4.3. Clinical Study

#### 4.3.1. Subjects

Twenty-one healthy female adults aged 50–60 years with no skin diseases or allergies participated in this study after providing informed consent (average age 53.7 ± 2.72).

#### 4.3.2. Formulation of Skin Care Cream

The vehicle cream was formulated as an O/W cream consisting of water, glycerin, caprylic/capric triglyceride, cetearyl alcohol, dimethicone, 1,2-hexanediol, cetearyl olivate, sorbitan olivate, carbomer, tromethamine, ethylhexylglycerin, and citric acid to adjust the pH to 7.0. Skin care cream containing NHE1 activator for clinical study was formulated by mixing 0.05% *M. officinalis* leaf extract (Maruzen Pharmaceuticals Co., LTD. Hiroshima, Japan) containing RA at more than 0.03%, and 0.1% rosmarinic acid (Avention, Incheon, Korea) with vehicle cream.

#### 4.3.3. Procedures

Skin care creams containing NHE1 activator (Cream A) and vehicle cream (Cream B) were applied twice daily (in the morning and evening) on each forearm for 4 weeks. Skin surface pH, TEWL, and skin hydration were evaluated at 0, 2, and 4 weeks after cream application; untreated sites were also evaluated.

After measuring pH, TEWL, and skin hydration, the same sites on each forearm were damaged by a patch containing 1% SLS for 24 h. The pH was measured before application of SLS patch (i.e., before damage); immediately after patch removal (i.e., immediately after damage); and 1, 3, and 7 days after patch removal (i.e., 1, 3, and 7 days after damage, respectively); sites without cream application were also evaluated. Creams A and B were not applied after SLS damage.

The skin surface pH was measured using a Skin-pH-Meter 905 (Courage + Khazaka Electronic, Köln, Germany), and SC hydration was measured using a Corneometer (CM 825, Courage + Khazaka Electronic). TEWL was measured using a Tewameter (TM 300, Courage & Khazaka Electronic). Prior to all measurements, subjects remained in the room for at least 30 min to allow full skin adaptation to room temperature (20–24 °C) and humidity (40–60%). Clinical study was conducted at the SKINMED Clinical Trials Center.

#### 4.3.4. Statistical Analysis

All statistical analyses, including calculation of mean and standard deviation values, were performed using SPSS™ Statistics 26 (IBM, Armonk, NY, USA). Conformity with normal distribution was determined using the Kolmogorov–Smirnov test, following which parametric repeated measures ANOVA and independent samples *t*-test and nonparametric Mann–Whitney U-Test and Wilcoxon signed-rank *t*-test were performed. Statistical significance was set at *p* < 0.05.

## 5. Conclusions

In conclusion, NHE1 activators, such as rosmarinic acid, decreased skin surface pH and significantly improved skin barrier functions. This was validated by increased ceramide formation, lower TEWL, and elevated skin hydration. Recently, weakly acidic topical applications have been developed. Considering the results of our study, the application of creams containing NHE1 activators, such as rosmarinic acid, can potentially be a more effective therapeutic strategy compared to simple acidic topical applications. Our results further the rationale for adding NHE1 activators in topical applications to improve skin barrier functions.

## Figures and Tables

**Figure 1 ijms-23-03910-f001:**
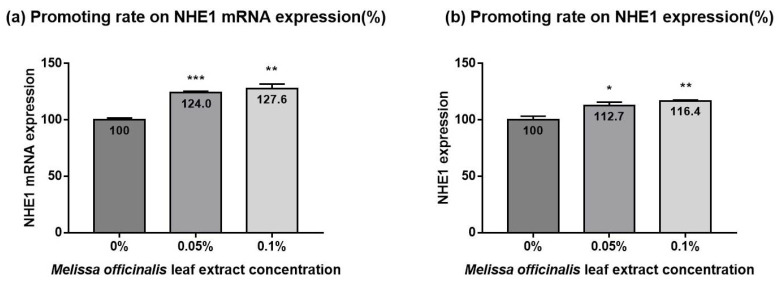
Promoting effects of *Melissa officinalis* leaf extract on NHE1 expression. *M. officinalis* leaf extract-containing solution increased expression levels of NHE1 mRNA (**a**) and NHE1 (**b**), compared to the control. Data are presented as the mean ± SD (*n* = 3). NHE1, sodium proton exchanger 1. Student’s *t*-test *p*-values indicate statistical significance (* *p* < 0.05, ** *p* < 0.01, and *** *p* < 0.001).

**Figure 2 ijms-23-03910-f002:**
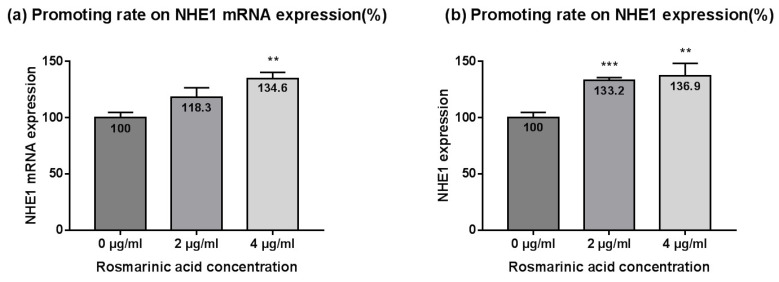
Promoting effects of rosmarinic acid on NHE1 expression. Rosmarinic acid-containing solution increased expression levels of NHE1 mRNA (**a**) and NHE1 (**b**), compared to the control. Data are presented as the mean ± SD (*n* = 3). NHE1, sodium proton exchanger 1. Student’s *t*-test *p*-values indicate statistical significance (** *p* < 0.01, and *** *p* < 0.001).

**Figure 3 ijms-23-03910-f003:**
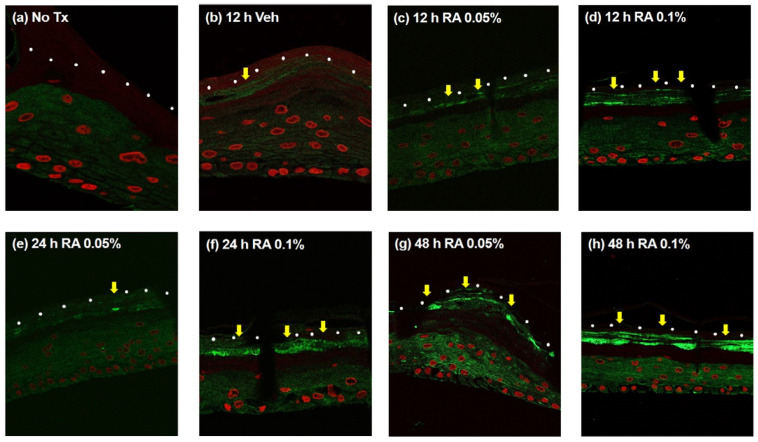
Immunofluorescence (IF) staining for analyzing NHE1 expression in 3D-cultured skin after topical application. IF staining for NHE1 expression after application of the NHE1 activator cream (containing 0.05% or 0.1% rosmarinic acid) or the vehicle cream. (**a**) No treatment, (**b**) 12 h after vehicle cream application, (**c**) 12 h after rosmarinic acid 0.05% cream application, (**d**) 12 h after rosmarinic acid 0.1% cream application, (**e**) 24 h after rosmarinic acid 0.05% cream application, (**f**) 24 h after rosmarinic acid 0.1% cream application, (**g**) 48 h after rosmarinic acid 0.05% cream application, and (**h**) 48 h after rosmarinic acid 0.1% cream application. The yellow arrows indicate NHE1 expression between the SC and SG. Red, nuclei of keratinocytes; white dots, interface between SC and SG. RA, rosmarinic acid; Veh, vehicle.

**Figure 4 ijms-23-03910-f004:**
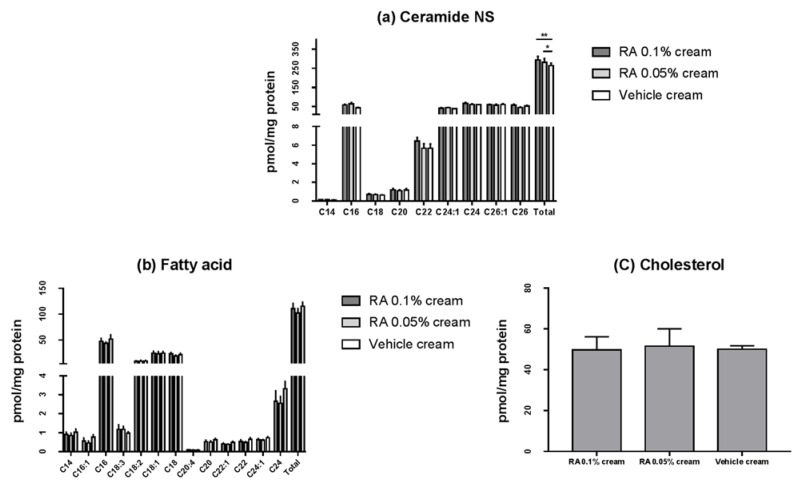
Epidermal lipid analysis in 3D-cultured skin after topical application. Epidermal lipid analysis after the NHE1 activator cream (containing 0.05% or 0.1% rosmarinic acid) or vehicle cream application. (**a**) Ceramide NS, (**b**) fatty acid, and (**c**) cholesterol. Data are presented as the mean ± SD (*n* = 3). RA, rosmarinic acid. Unpaired Student’s *t*-test *p*-values indicate statistical significance (* *p* < 0.05, and ** *p* < 0.01).

**Figure 5 ijms-23-03910-f005:**
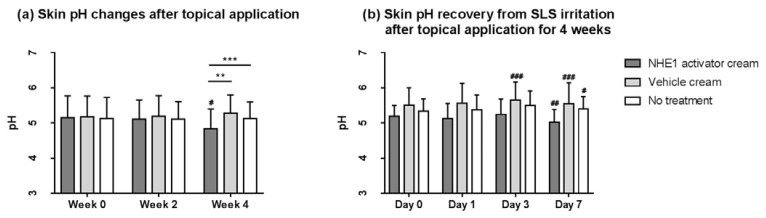
Skin pH changes after the NHE1 activator cream application. (**a**) Skin pH was measured after the application of the NHE1 activator cream (*M. officinalis* leaf extract 0.05% and rosmarinic acid 0.1%) on the forearm, and application of the vehicle cream or no treatment on the other forearm. (**b**) After topical applications (NHE1 activator cream or vehicle cream) or no treatment for 4 weeks, skin pH recovery after sodium lauryl sulphate (SLS) irritation was assessed. Data are presented as the mean ± SD (*n* = 21). Wilcoxon signed-rank *t*-test and Mann–Whitney U test *p*-values indicate statistical significance (** *p* < 0.01, and *** *p* < 0.001; # *p* < 0.05, ## *p* < 0.01, and ### *p* < 0.001).

**Figure 6 ijms-23-03910-f006:**
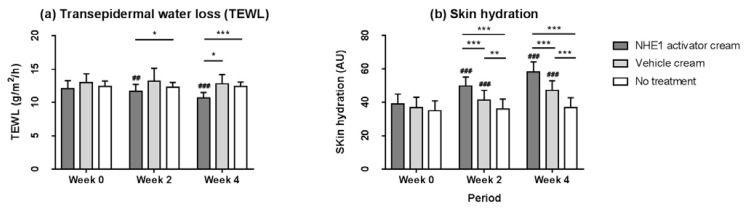
Changes in transepidermal water loss (TEWL) and skin hydration after application of the NHE1 activator cream or vehicle cream. (**a**) TEWL was measured after application of the NHE1 activator cream (*M. officinalis* leaf extract 0.05% and rosmarinic acid 0.1%) on the forearm and vehicle cream on the other forearm. (**b**) After topical applications (NHE1 activator cream or vehicle cream), skin hydration was assessed. Data are presented as the mean ± SD (*n* = 21). Repeated measures ANOVA, independent samples *t*-test, Wilcoxon signed-rank *t*-test, and Mann–Whitney U test *p*-values indicate statistical significance (* *p* < 0.05, ** *p* < 0.01, and *** *p* < 0.001; ## *p* < 0.01, and ### *p* < 0.001).

## Data Availability

The datasets generated and/or analyzed during the current study are available from the corresponding author, Eung Ho Choi, upon reasonable request.

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
