# Peer review of "Rosmarinic Acid, as an NHE1 Activator, Decreases Skin Surface pH and Improves the Skin Barrier Function"

_ijms, 2022, doi:10.3390/ijms23073910_

Round 1

Reviewer 1 Report

This study reported the finding that rosmarinic acid, a pure compound of Melissa officinalis leaf extract, could increase the expression of NHE1 in cultured keratinocytes and 3-D cultured skin.  Rosmarinic acid was found to increase the level of cermamide, but not fatty acid and cholesterol in 3-D cultured skin. Cream containing rosmarinic acid decreased pH and transepidermal water loss in human skin. The skin hydration was also found to be improved.

Comments to this study--

  1. Author names the cream as a NHE1 activator-containing cream, however, the activation of NHE1 function has not been proved in this manuscript.
  2. The expression of NHE1 was only determined by ELISA that was not enough since NHE1 is a membrane protein, at least the expression of NHE1 on membrane is needed to be proved.
  3. In Fig. 3, there is no positive staining of nucleus colocalized with the green fluorescence. Author needs to explain the reason or better pictures need to be shown.
  4. In Fig. 4, the difference of ceramide level is too small to prove the effect of rosmarinic acid.
  5. It is better to show the fingerprint of Melissa officinalis leaf extract for readers to understand the proportion of rosmarinic acid.
  6. The critical role of NHE1 in skin barrier function needs more experiments to demonstrate such as by knockdowning NHE1 and then analyzing the effect on related skin barrier functions.
  7. Author needs to explain why in human, 4 weeks is required to see the effect of rosmarinic acid that is much longer than 2 days for the 3-D cultured skin.

Reviewer 2 Report

The paper "Rosmarinic Acid, as an NHE1 Activator, Decreases Skin Surface pH and Improves Skin Barrier Function" examines the influence of both a Melissa officinalis extract and also of the proposed primary active from the extract, Rosmarinic Acid, on skin pH and skin barrier function.  The paper offers a good combination of in vitro studies examining the effects of the extract and active on NHE1 expression and a sufficiently well-done clinical study to support pH and barrier function improvements.  This referee noted that there was no attached Ethical Declaration stating that the clinical work was done in accordance with appropriate control under Helsinki.  That should be included.

The paper appears to be missing some key references including 31- 36 (Lines 332, 342 and 350).  The references that this referee checked were accurate and correct.  

The paper is well-written and clear and the studies flow nicely from the in vitro work to the clinical work.  The conclusions are sound and the data is presented well.  It was noted that there was a significant number of papers cited by Peter Elias et al.  This is appropriate given his work in the area, but there may be other critical authors who have examined the influence of pH on skin barrier that might be considered.  This is not a critical problem, however.

The primary problem this referee has with the paper is that the initial in vitro work was done with the Melissa officinalis extract.  The authors then moved to examining the active, Rosmarinic Acid, without making any direct comparison of the amount of active in the extract verses the levels of Rosmarinic acid then tested.  Does the extract contain appreciable levels of Rosmarinic Acid or not?  The authors should provide some idea how much Rosmarinic Acid is being delivered from the extract as there seems to be little direct correlation between their extract results and their active results.  This becomes particularly important in the clinical work where the formulations then include both the extract and the Rosmarinic acid!  Again, is there a significant contribution of Rosmarinic Acid from the extract or not?  This is important. 

If the authors can address the deficiencies above, the paper is suitable for publication.      

Round 2

Reviewer 1 Report

The author did not properly answer the questions raised by reviewer.  There were  no additional experimental data to further support the conclusion made by author either.

Author Response

We would like to thank reviewer for the critical and constructive reviews to supplement our manuscript. Unfortunately, we are unabled to conduct further experiments at the moment, and it is a disappointing that we could not answer with additional data. However, we will reflect your comments on further research and supplement our research.